# Alcohol reduction outcomes following brief counseling among adults with HIV in Zambia: A sequential mixed methods study

Mah Asombang[1]☉*, Anna Helova[2]☉, Jenala Chipungu[1], Anjali Sharma[1], Gilles Wandeler[3], Jeremy C. Kane[4], Janet M. Turan[2], Helen Smith[5], Michael J. Vinikoor[1,6], for IeDEA Southern Africa

**1** Centre for Infectious Disease Research in Zambia, Lusaka, Zambia, **2** School of Public Health, University of Alabama at Birmingham, Birmingham, AL, United States of America, **3** Department of Infectious Diseases and Institute of Social and Preventive Medicine, University of Bern, Bern, Switzerland, **4** Columbia University Mailman School of Public Health, New York City, NY, United States of America, **5** Bradford Institute for Health Research, Bradford, United Kingdom, **6** School of Medicine, University of Alabama at Birmingham, Birmingham, AL, United States of America

☉ These authors contributed equally to this work.
* Mah.Asombang@cidrz.org

**Data Availability Statement:** To ensure adherence to Zambian research laws, data from this study are available only by request to the Centre for

## Abstract

Data from sub-Saharan Africa on the impact of alcohol on the HIV epidemic in sub-Saharan Africa is limited. In this region, it is not well understood how people with HIV (PLWHA) respond to alcohol reduction counseling while they are linked to HIV clinical care. We conducted an explanatory sequential mixed-methods study to understand patterns of alcohol use among adults (18+ years) within a prospective HIV cohort at two urban public-sector clinics in Zambia. At antiretroviral therapy (ART) start and one year later, we measured alcohol use with Alcohol Use Disorders Identification Test-Consumption (AUDIT-C) and those reporting any alcohol use were provided brief counseling. We conducted focus groups at 1 year with participants who had any alcohol use and 20 in-depth interviews among the subgroup with unhealthy use pre-ART and who either reduced or did not reduce their use by 1 year to moderate levels or abstinence. Focus group Discussions (FGDs) (n = 2) were also held with HIV clinic staff. Qualitative data were analyzed using thematic analysis. The data obtained from 693 participants was analyzed (median age 34 years, 45% men), it revealed that unhealthy alcohol use (AUDIT-C >3 for men; >2 for women) was reported among 280 (40.4%) at baseline and 205 (29.6%) at 1 year on ART. Reduction from unhealthy to moderate use or abstinence was more common with older age, female, non-smoking, and at Clinic B (all P<0.05). Qualitative data revealed ineffective alcohol support at clinics, social pressures in the community to consume alcohol, and unaddressed drivers of alcohol use including poverty, poor health status, depression, and HIV stigma. Healthcare workers reported a lack of training in alcohol screening and treatment, which led to mixed messages provided to patients ('reduce to safe levels' versus 'abstain'). In summary, interventions to reduce unhealthy alcohol use are needed within HIV clinics in Zambia as a substantial population have persistent unhealthy use despite current HIV clinical care. A better understanding is

Infectious Disease Research in Zambia data analysis unit (samuel.bosomprah@cidrz.org).

**Funding:** This research was supported by the National Institute of Allergy and Infectious Diseases under grant number U01AI069924 (GW and MJV), the National Institute of Alcohol Abuse and Alcoholism under grant numbers K01AA026523 (JCK) and P01AA029540 (AS, JCK, and MJV), and the Fogarty International Center at the United States National Institutes of Health under grant number K01TW009998 (MJV). Additional support came from the University of Alabama (UAB) Center for AIDS Research P30 AI027767 (MJV) and Sparkman Center for Global Health (AH and JT). The funders had no role in study design, data collection and analysis, decision to publish, or preparation of the manuscript.

**Competing interests:** The authors have declared that no competing interests exist. Authors Jenala Chipungu and Michael Vinikoor were unavailable to confirm their authorship contributions. On their behalf, the corresponding author has reported their contributions to the best of their knowledge.

needed regarding the implementation challenges related to screening for unhealthy alcohol use integrated with HIV services.

## Introduction

Excessive unhealthy alcohol consumption is common in Zambia, particularly among persons living with HIV/AIDS (PLWHA) [1, 2]. The negative consequences of excessive alcohol consumption are enhanced in PLWHA due to their pre-existing immunocompromised state and increased risk of transmitting HIV to other people and acquiring opportunistic infections [3–5]. Furthermore, alcohol use among this population delays HIV diagnosis and can undermine adherence to antiretroviral (ARV) treatment) [6, 7]. Unhealthy alcohol use also adversely affects sustained engagement in HIV care. Ultimately alcohol use is associated with higher morbidity and mortality in PLWHA) [6, 7]. Concerningly, when compared to the general population, PLWHA are also more likely to use alcohol [6, 8].

Despite its negative impact on the HIV care continuum, unhealthy alcohol use remains largely unaddressed in sub-Saharan Africa, as evidence-based interventions to reduce alcohol use are rarely implemented [6, 7, 9]. A systematic review found only 14 studies that used an alcohol-reduction intervention to prevent HIV acquisition in Africa and the results of these studies were mixed [9]. Most interventions in this area have been brief in nature; however, a pilot study in Kenya demonstrated the feasibility of a multi-session alcohol reduction intervention based on cognitive behavioral therapy approaches [10]. As data evidence and understanding improves around what helps PLWHA reduce alcohol intake in Sub-Saharan Africa it will be important to examine the barriers and obstacles to providing alcohol reduction interventions in the Zambian clinical context. This will provide better understanding of patient and system-level factors that are associated with HIV outcomes and help identify system and service-level improvements.

This study aimed to better understand alcohol consumption practices and determinants among PLWHA in urban Zambia and to characterize gaps in the management of unhealthy alcohol use at public HIV clinics in Zambia. The findings of this study will provide guidance for the Zambian Ministry of Health (MOH) to provide evidence-based alcohol reduction programs during standard HIV care.

## Methods

In a liver disease-focused prospective cohort study of adults with HIV in urban Zambia [11], we identified a high prevalence of unhealthy alcohol use that persisted after initiation of ART and with receipt of counseling that touched on alcohol use. To better understand patterns of unhealthy alcohol use within the cohort, particularly the failure of many patients to curtail their drinking, we conducted a sequential explanatory mixed-method evaluation (see Fig 1) [12]. Our approach in study design, content of interview/focus group guides, and interpretation of data was guided by the adapted Andersen's Behavioral Model [17], which suggests that determinants of health behaviors include the healthcare environment, patient characteristics, the community, the societal context, and patient's need for health services [13]. The overall cohort and the protocol amendment that added new data necessary for the mixed methods evaluation were approved by the University of Zambia Biomedical Research Ethics Committee, Lusaka, Zambia, and the University of Alabama at Birmingham Institutional Review Board.

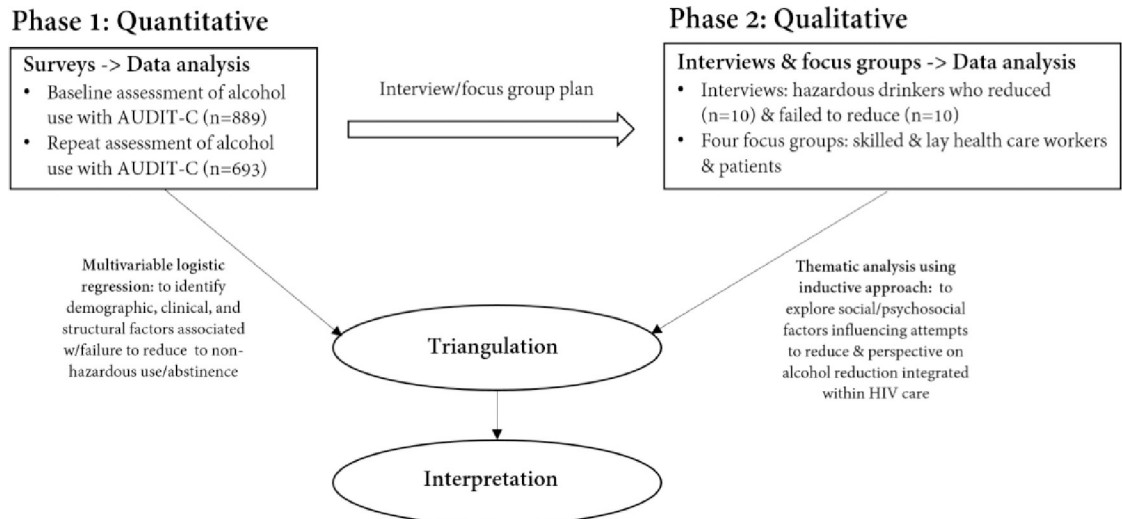

**Fig 1. Process flow diagram of the study procedures.** Explored mixed method research design.

## Quantitative methods

The IeDEA Hepatitis cohort used in this study was previously described [11]. Patients at 2 urban clinics in Lusaka were enrolled consecutively (up to 8 enrollments per day) if they gave consent and met the following inclusion criteria: age ≥ 18 years old, HIV-positive status, and initiating ART at a study site. During a later phase of enrollment, we narrowed inclusion to patients who also had hepatitis B coinfection (based on hepatitis B surface antigen-positivity), a focus of the cohort. Patients were excluded if they had intentions of relocating to another facility.

Alcohol use was measured during enrollment while the patient was being initiated on ART, and at 1 year later using the 3-question Alcohol Use Disorders Identification Test-Consumption (AUDIT-C) [14, 15]. AUDIT-C measures were implemented the same way at the two sites. The enrollment clinic (both were large urban sites with 5,000–10,000 PLWHA in care) was noted in each patient's record to allow for analysis of clinic-level differences. However, we did not measure staffing levels, competency of staff, and to what degree the staff emphasized management of alcohol-related issues.

Participants received routine ART adherence counseling at each visit. When alcohol use was reported, this counseling included brief and unstructured information on reducing alcohol consumption. Alcohol reduction may mean alcohol content in a drink or body is reduced. Patients also received general information on the potential harms of alcohol use during 'health talks' delivered by counselors to patients waiting in the queue for ART services.

Analysis focused on AUDIT-C data from baseline and the 1-year visit; thus, patients who were not retained to 1 year, because of death, transfer out, or loss to follow-up (LTFU), were excluded from analysis. Among patients with unhealthy use at baseline, we described the mean and standard deviation change in AUDITC score, from baseline to one year, as well as the proportion who achieved alcohol reduction, which we defined as transition to moderate use or abstinence at one year. The WHO recommends the full AUDIT-C and proposes thresholds [15]. We defined moderate alcohol use as 1–2 points for women and 1–3 points for men, while higher points defined unhealthy alcohol use and abstinence was 0 points based on multiple research studies [16]. We used a multivariable logistic regression on the data collected from participants with unhealthy alcohol use at enrollment. We examined independent factors

potentially associated with alcohol reduction from the literature, including age, sex, marital status, education, WHO clinical stage, smoking, baseline CD4+ count, and clinic. A stepwise logistic regression model was used to identify factors associated with attainment of alcohol reduction at one year using a forward selection algorithm. The probability of selection was set at 0.2 using a likelihood ratio test.

## Qualitative methods

To explore their perspectives, beliefs, and norms surrounding alcohol consumption in relation to HIV infection and care received at the clinic we facilitated two focus group discussions (FGDs) with people living with HIV and reporting unhealthy alcohol use (one with men and one with women). To understanding perspectives around alcohol reduction in these individuals, we also conducted twenty in-depth interviews (IDIs) with participants based on their patterns of alcohol use from the AUDIT-C. We included ten (five men and five women) who reported unhealthy use at baseline and moderate use or abstinence at 1 year and ten (five men and five women) who reported unhealthy alcohol use at both baseline and 1 year. Finally, we held two FGDs with lay and professional healthcare workers (HCW) providing services at the recruitment sites to gather additional information about the healthcare environment in relation to HIV and alcohol treatment.

Potential participants in the qualitative phase were contacted by a research team member and if agreeable, informed consent (for patients, this was an additional consent on top of the parent cohort consent) was obtained prior to qualitative data collection. Patients were allowed to participate in both an FGD and IDI. All FGDs and IDIs were conducted by a trained research assistant or investigator (J.C.) in private settings using a mixture of English and local Zambian languages. Participant characteristics from the quantitative data were linked to the qualitative data. FGDs and IDIs were digitally recorded, translated to English when necessary, and transcribed verbatim, excluding any identifying information. All files were password protected and stored in a secure location. Subsequently, transcripts were coded using the Dedoose qualitative software program (Sociocultural Research Consultants, LLC). Coding and analysis followed a reflexive thematic analysis approach, in which broad coding framework precedes theme development [17, 18]. The broad coding framework was based on the literature, constructs of the Andersen's Behavioral Model, topics from interview guide, and in the later phase of coding, based on emerging themes inductively derived from transcripts. Transcripts were initially broad coded by three individuals trained in qualitative coding, and consistency of coding was established. Excerpts from broad codes were then fine coded using an inductive approach.

**Ethics statement.** Ethical approvals for the quantitative and qualitative parts of the study were obtained from the University of Zambia Biomedical Research Ethics Committee and the University of Alabama at Birmingham Institutional Review Board.

**Patient consent.** Written informed consent was obtained from all participants.

## Results

### Quantitative results

From October 2013 to September 2015, 897 patients were enrolled (797 during the initial phase and an additional 100 with HBV coinfection). The present analysis included 693 (77.2%) cohort patients with non-missing AUDIT-C data at enrollment and at 1 year. A total of 204 participants were not included because of missing AUDIT-C data(15), loss to follow-up (n = 102), death (n = 57), or transfer out (n = 30) prior to 1 year. Age, sex, CD4+ count, and unhealthy alcohol use were similar between cohort participants included and excluded from

**Table 1. Baseline characteristics of 693 PLWHA in Lusaka, Zambia, by alcohol use at baseline.**

| | Abstinent or moderate alcohol use (n = 413) | Unhealthy alcohol use (n = 280) | P* |
|---|---|---|---|
| Age, years | | | |
| 18–29 | 110 (26.6) | 73 (26.1) | 0.05 |
| 30–39 | 176 (42.6) | 142 (50.7) | |
| 40 and above | 127 (30.8) | 65 (23.2) | |
| Sex | | | |
| Men | 135 (32.7) | 179 (63.9) | <0.01 |
| Women | 278 (67.3) | 101 (36.1) | |
| Education level | | | 0.01 |
| None or primary | 320 (78.4) | 194 (69.8) | |
| At least secondary | 88 (21.6) | 84 (30.2) | |
| Marital status | | | |
| Never married | 25 (6.1) | 33 (11.8) | 0.007 |
| Married or cohabiting | 258 (63.2) | 182 (65.2) | |
| Divorced or widowed | 125 (30.6) | 64 (22.9) | |
| WHO stage | | | 0.84 |
| 1 or 2 | 240 (58.4) | 159 (57.6) | |
| 3 or 4 | 171 (41.6) | 117 (42.4) | |
| HBV coinfection | | | |
| No | 355 (86.0) | 214 (76.4) | 0.001 |
| Yes | 58 (14.0) | 66 (23.6) | |
| Cigarette smoking | | | |
| No | 403 (97.6) | 214 (76.4) | <0.01 |
| Yes | 10 (2.4) | 66 (23.6) | |
| CD4+ count, cells/mm$^3$ | 225 (122–338) | 241 (128–347) | 0.33 |
| Clinic | | | |
| A | 261 (63.2) | 179 (63.9) | 0.84 |
| B | 152 (36.8) | 101 (36.1) | |

All values are median (interquartile range) or number (%).

*Comparisons were by Chi square for categorical variables and Wilcoxon rank sum for CD4+ count.

Abbreviations: PLWHA, persons living with HIV; ART, antiretroviral therapy; WHO, World Health Organization; HBV, hepatitis B virus

this analysis (P>0.05). Characteristics of study participants included in the present analysis are summarized in Table 1. Within the analysis cohort, median age at enrollment was 34.9 years, 379 (54.8%) were women, and median baseline CD4 count was 234 cells/mm$^3$. Unhealthy alcohol use was reported at baseline by 280 patients including 36.1% of women and 63.9% of men (P<0.01).

During the first year on ART the overall prevalence of unhealthy alcohol use in the analysis group reduced significantly from 40.4% to 29.6% (P<0.01). Of the 280 with unhealthy use at baseline, 122 (43.6%) reported a lower degree of alcohol consumption at 1 year. This was partially offset by an additional group of 47 of 413 (11.4%) who reported abstinence or moderate use at ART start and then reported unhealthy alcohol use at their 1-year visit. In multivariable analysis of those with baseline unhealthy use, factors associated with reduced alcohol use were younger age, female sex, non-smoking status, and clinic (Table 2). Compared to 18–29 years old, 30–39 years (adjusted odds ratio [AOR], 0.49; 95% confidence interval [CI], 0.26–0.92) had lower odds of alcohol reduction. Men (AOR, 0.41; 95% CI, 0.23–0.72) and baseline smokers (AOR, 0.49; 95% CI, 0.25–0.96) also had lower odds of reducing their alcohol use after

**Table 2. Factors associated with alcohol reduction after ART initiation among PLWHA with a history of unhealthy alcohol use (N = 280).**

|  | Crude OR (95% CI) | Adjusted OR (95% CI) * |
|---|---|---|
| Age, years |  |  |
| 18–29 | Reference | Reference |
| 30–39 | 0.43 (0.24–0.76) | 0.49 (0.26–0.92) |
| 40 and above | 1.38 (0.70–2.72) | 1.78 (0.86–3.70) |
| Male sex | 0.34 (0.20–0.56) | 0.41 (0.23–0.72) |
| At least secondary education | 1.36 (0.81–2.27) |  |
| Marital status |  |  |
| Never married | Reference |  |
| Married or cohabiting | 1.34 (0.62–2.89) |  |
| Divorced or widowed | 1.64 (0.69–3.89) |  |
| WHO stage 3 or 4 | 1.01 (0.62–1.63) |  |
| HBV coinfection | 0.73 (0.42–1.29) |  |
| Cigarette smoking | 0.40 (0.22–0.73) | 0.49 (0.25–0.96) |
| Baseline CD4+, per 50-cell increase | 0.96 (0.88–1.04) |  |
| Clinic B | 2.42 (1.47–4.00) | 2.92 (1.69–5.04) |

*Factors associated with outcome at P<0.2 during bivariable analysis were included in multivariable analysis.

Abbreviations: ART, antiretroviral therapy; OR, odds ratio; CI, confidence interval; BMI, body mass index; WHO, World Health Organization; HBV, hepatitis B virus

ART commencement. There was also increased odds of alcohol reduction at Clinic B (AOR, 2.91; 95% CI, 1.695.04) compared to Clinic A.

## Qualitative results

The main themes resulting from the qualitative analysis were related to drinking norms and generational changes, motives for alcohol drinking among PLWHA, drinking alcohol and poor HIV adherence, and insufficient support available at HIV clinics to reduce alcohol consumption.

**1) Drinking as a cultural norm and changing generational normative behavior.** According to FGD participants, alcohol consumption is a societal and cultural norm in urban Zambia.

> *"People here drink a lot of alcohol. [. . .] I don't feel anything [about being heavy drinker] because there are a lot of people who drink alcohol."*–Female PLWHA (FLWHA), unhealthy drinking

The majority defined 'unhealthy' use as consuming alcohol every day and/or consuming certain "harder" forms of alcohol, e.g. junta–whiskey, Kachasu–an illegal traditional distilled beverage, and tujilijili—the banned highly intoxicating illicit alcohol packed in sachets. This was described as drinking that resulted in inability to perform daily responsibilities, and possibly violence.

> *"I see how a lot of people get drunk, even in bars where we are found. When a person is very drunk, just right there he will pick up a fight, they will start insulting. Even when they are walking, they will start to lose balance as they walk [. . .]. He can't think of anything or just*

*think about this same disease [HIV] [. . .], they forget everything they even fail to eat.*"–Male PLWHA (MLWHA), reduced drinking

Older participants frequently expressed disapproval of unhealthy alcohol use among women, particularly if they are married with children, and unable to fulfill family obligations.

*"Drinking moderately like you have said, drinking moderately like for women, we are talking about women not us men, we are different. [. . .] A woman is not supposed to take alcohol and it's not good. [. . .] We will find that women have gone to their parties, kitchen parties and so forth, they have the right to drink but they have to drink reasonably. Not drink [such] that every day she is coming home in the morning, failing to take care of her children and forgetting. No, drinking is normal, but not drinking excessively."*–MLWHA (FGD)

Generational and societal changes, including increased unhealthy alcohol consumption among young people were noted. Participants also reported that alcohol use among women was becoming more normative than it had been historically.

*"I don't know if its life or they are trying to show that they are clever when they are drinking alcohol. Believe you me, children who are less than 20, they are already in the bars. [. . .] When you look backwards, you will realize that these things never use to happen. Why? Women are no longer shy, I mean life has taken them."*- Lay HCW (FGD)

**2) Motives for alcohol drinking among PLWHA.** Most patients thought that PLWHA drank alcohol in order to cope with the many demands and sources of anxiety and stress in their lives, including their HIV diagnosis, other health challenges, relationship issues, being single or widowed, and being poor.

*"It [drinking] just helps me avoid a lot of thoughts. I don't get deep into the thoughts such that I start telling myself that I am sick or even starting to think of hanging myself because of HIV. So, in other words, it makes me to open up and feel to be the same [as before HIV diagnosis]."*–MLWHA, reduced drinking

Some participants said that their alcohol intake sometimes increased following HIV diagnosis. Patients shared that one of the main reasons for persistent unhealthy alcohol consumption, despite awareness of potential harms, was to prevent inadvertent disclosure of their HIV status, inferred by their drinking friends if they suddenly changed their behavior, i.e., stopped taking alcohol.

*"The time when I drink the most is when people who don't know my status. . . come to visit me and you find that when these people have brought alcohol. How am I going to refuse?! Because those people know that I drink alcohol and I used to drink a lot and then they find that I am not drinking, how will it be? So those are times when I take even 4 bottles and slowly, slowly drink."*–FLWHA, reduced drinking

PLWHA also feared stigmas associated with HIV and heavy drinking. Many participants expressed willingness to reduce their alcohol intake; however, they found it difficult to fight addiction, particularly when facing the challenge with limited support.

*"I would want to stop, okay I would want. There are times when I can even go three days without drinking alcohol but the kind of thirst I will have will be different from that of wanting to drink water."*–MLWHA, unhealthy drinking

Participation and not drinking during social/community events and gatherings (e.g. kitchen parties, weddings, funerals, spending time with friends/family members, watching sports) were presented as one of the most challenging barriers to achieving alcohol abstinence and reduction. Some simply stated that drinking is their source of enjoyment which is difficult to give up.

*"It's the group influence which made me start using alcohol. When you are in the group, they will be telling you that "Have you seen this one? He is also taking the drugs (ARVs). There is no problem [with drinking when on ARVs], you just have to drink alcohol."*–MLWHA, reduced drinking

**3) Drinking alcohol and poor HIV treatment adherence.** Perception of what reduced alcohol use entailed varied among study participants and included lower frequency of drinking, reduced number of alcohol or other substances consumed (e.g., crushed tobacco), and switching from stronger forms of alcohol like spirits to 'acceptable' alcohol types like beer. "Chibuku" or "Shake Shake" (a traditional-style beer that is now made commercially from maize/sorghum and was referred to as 'food alcohol' or 'dinner and drinks'). While on ART, these traditional beers were an accepted norm, often considered beneficial, and sometimes recommended by HCWs as a safe alternative to 'hard alcohol' for patients with HIV on treatment.

*"When they were teaching us here at the clinic, they said that if you take medicines (ARVs), you are not supposed to drink alcohol. But we people, we tend to fail, you see and that is why we have chosen something which is much better. Okay on my side, I have chosen to drink Shake Shake."*–MLWHA, unhealthy drinking

Most patients believed that if the amount of alcohol intake is controlled, it would not cause problems with ART adherence or retention in care; however, unhealthy alcohol use could reduce ART adherence and retention in care in multiple ways. They mentioned that alcohol impaired the mental resources needed to adhere to HIV treatment, including the ability to keep clinic appointments, particularly in the morning hours. Some held the belief that ARVs and alcohol should not be mixed; and therefore, did not take ARV pills on days of alcohol consumption. Others held the belief that a person could not both engage in frequent alcohol consumption and adhere to HIV treatment.

*"Those who drink, sometimes they even know that they are supposed to come and collect the medicine [ARVs]; they will then pass through the bar and forget to come and that's it! He would remember the next day and say "Let me go." Some come [to clinic], and others just drink alcohol."*–FLWHA, reduced drinking

**4) Insufficient support available at HIV clinics to reduce alcohol consumption.** Anticipated and experienced stigma, harsh treatment, shouting, and 'punishment' at the HIV clinic due to alcohol use were discussed by PLWHA. Participants consequently felt they had received poor HIV care services, which made them less likely to return for services. Patients reported

being penalized by HCWs for missing their clinic appointments or not arriving to the clinic early in the morning (a cultural norm in Zambia) because of drinking.

> *"They know the implications, they know side effects, that's why that basic thing of telling lies thinking that "Oh! If I say the truth, they will take me to sister XX for weekly adherence and they will start monitoring very closely." So, they would do their best [to avoid this and] tell lies. [. . .] So, it's up to you now to look in his or her eyes and say, "Just tell me the truth." that's when somebody will reveal [that], "Ahhh! Sorry doctor, I did drink just a few."*

Overwhelmingly, patients expressed a lack of support with efforts to reduce their alcohol use beyond the brief and inconsistent counseling they received in the ART clinic. Both HCWs and patients reported limited alcohol-intervention related resources, lack of community-based resources, lack of counselling services targeting PLWHA, and lack of support groups or peer mentors. HCWs also reported that they had received virtually no training focused on how to diagnose and counsel PLWHA with unhealthy alcohol use and mental health comorbidity, expressing concern about the reliance on patient self-reports to detect unhealthy drinking.

> *"I: Mainly we just interview them [patients], ask them if they do drink, how much they drink, and how often they drink.*
>
> *I: So how accurate is this assessment?*
>
> *P: It's not accurate as such [. . .] but that's the means, that's the only way we do. If there are relatives, we will also try to find out maybe from the wife or caregivers, "Do they drink alcohol? How much do they drink?"*—Professional HCW (FGD)

Both patients and HCWs discussed that while some patients truthfully disclose their drinking, others do not. Some reasons included shame, fear of differential treatment from HCWs, fear of inadvertent disclosure of drinking to other people present at the clinic and being perceived as 'difficult' due to heavy drinking, and a fear that they might be subjected to weekly adherence meetings and closer monitoring–which they felt was punishment.

> *"I: So, do those that come to the hospital tell the staff that, "Yes, me, I drink?"*
>
> *P: No, they don't come out. [. . .] They feel shy, thinking "Aaah! I reveal to the crowd that*
>
> *I drink?!" There are a lot of people, madam! Were you not there in the morning? There are a lot of people and then you just come out with "Me, I drink? Aaaa! No you can't! It's embarrassing!"*–MLWHA, reduced drinking

HCWs reported using various approaches to counsel patients to reduce their alcohol intake. Some HCWs advocated for immediate abstinence, while others recommended reduction without abstinence and a shift to drinking only 'safer' options (e.g., Shake Shake) and lower amounts of alcohol intake. Some HCWs chose to educate their patients on the harms of alcohol use and leave it up to the patient to decide to either reduce, stop, or continue at current drinking level.

> *"P: I tell them about the side effects, the interaction of ARVs and alcohol [. . .] You can't say that stop all of sudden, but you give them the information, then somebody will decide that it's not good to drink alcohol while you are taking the ARVs [. . .]. Upon listening to the*

*information you give them, they would weigh the risk and benefits. [. . .] So, in short you tell them to reduce but it's up to them to stop."*—Professional HCW (FGD)

Patients reported the counseling felt 'incomplete' at times without adequate explanation of benefits of reduced drinking or on how to reduce their use.

*"We went for testing and that was when I was found to be HIV positive with my wife. [. . .]*

*They said, "We have found you to be positive," and that is when they started asking us, "Do you take alcohol?" And that is when they said that you should stop taking alcohol, that is what they insisted—that we should stop. They didn't say that you should minimize or what, they said that you should stop taking alcohol and what I did is that I didn't manage to stop immediately, I just minimized to what I would manage."*–MLWHA, reduced drinking

Consistent with our quantitative findings, HCWs reported that women were more responsive to counseling to reduce alcohol use compared to men.

*"Women are easier to change than men. Even when people are coming to ART [clinic], we are seeing more women than men. Why? It's because women [..] when they hear things they will always think of their children, they say, "Ahhh! Let me look after my children, I don't want them to suffer". If you ask them where your husband is, they will say. "Ahhh, they don't want! I don't know but they don't want [to come to clinic]."*–Professional HCW (FGD)

Finally, a few patients reported they never received counseling to lower their alcohol use at the HIV clinic.

## Mixed-methods integration offindings

We used an adapted Andersen's Behavioral Model as a framework to identify key quantitative and qualitative findings related to unhealthy alcohol use among PLWHA in urban Zambia and the most frequently discussed barriers to alcohol reduction in the contextual environment at patient and healthcare levels (Table 3).

Contextually, alcohol consumption is common in Zambia. While society seems to be undergoing change in drinking norms, women who drink are often stigmatized, particularly if they are married with children and unable to perform their daily responsibilities. Older participants also reported excessive drinking in younger people as a generational change. At the individual level, unhealthy consumption of alcohol was common among PLWHA, particularly men, identified in both quantitative and qualitative data.

Younger people were also less likely to cut down on alcohol use after HIV diagnosis. The motivators to drink included the enjoyment of drinking, socializing, and peer pressure. The barriers to successful alcohol reduction among PLWHA included the inability to cope with the demands of daily life and poverty without drinking, the fear of inadvertent disclosure of HIV status if they refused to drink, particularly at social events; and, presence of mental health issues, including alcohol addiction, depression, and stress.

Many PLWHA perceived alcohol reduction as challenging and often discussed lack of family/social support if they decided to reduce. Many PLWHA with unhealthy alcohol intake before commencing ART continued to drink heavily. Intersectional stigma related to HIV and alcohol use were prevalent and negatively affected the health status of patients. At the healthcare system level (HIV clinics), alcohol-related interventions and resources were sorely lacking in the study health facilities and the level of support and type of services varied among clinics.

**Table 3. Barriers and facilitators of reduced alcohol use among adults with HIV/AIDS taking antiretroviral therapy in urban Zambia, based on adapted Andersen's Behavioral Model.**

| Factors | Quantitative results (Demographic, clinical, and structural data) | Qualitative results (Interviews & focus groups) |
|---|---|---|
| **Contextual Environment** | | |
| Drinking norms | Men more likely to consume alcohol than women (63. 9% men; 36.1% women). | Drinking is a societal and cultural norm |
| Gender norms | Women more likely to reduce alcohol consumption. | Generational changes in the roles of men and women |
| Drinking and violence impeded a functional daily life | Unknown prevalence of participants who encountered violence because of alcohol consumption. | Drinking was considered unhealthy if it impacted one's functional daily life and caused violence. |
| Culturally women who consumed alcohol were "judged" particularly if they were married with children. Especially if it impacted their ability to take care of their family. | Unknown percentage of women who experienced "judgement" because it was not measured. | Women who partook in alcohol consumption expressed that their spouses disliked their alcohol drinking habits. |
| Younger generation perceived to drink excessively | 18–29 year old age group (26.1%) with greater unhealthy alcohol consumption than 40 year old people (23.2%) | Older people with notion that younger people drank more though rationale was not disclosed. |
| **Individual Factors Among PLWHA** | | |
| Male sex | At baseline, men reported unhealthy alcohol use in comparison to women. | Younger people (18–29 years old) had lower disclosure of HIV status |
| Younger women reported unhealthy alcohol use | Men were 60% less likely to reduce recommended alcohol reduction (AOR = 0.41, CI 0.23–0.72) | Other motivators to drink were due to HIV stigma, stress relief, to socialize with friends, for enjoyment, lack of family and social support. |
| Poverty | Those smoking cigarettes were significantly less likely to reduce unhealthy alcohol use (AOR = 0.49, CI 0.25–0.96). | Poverty contributed to |
| Alcohol addiction, smoking, unhealthy alcohol use after ART commencement | Many patients discussed that they indulged in substance abuse compared to alcohol beverage drinking to cope with negative thoughts (AOR = 0.41, CI 0.23–0.72) | Unhealthy alcohol use perceived as a coping mechanism to address issues in life |
| Mental health issues | Significantly low perceived need to reduce alcohol consumption (AOR = 0.49, CI 0.25–0.96) | Women more likely to consume alcohol to avoid inadvertent stress |
| Poor health status | People with HBV co-infection less likely to partake in unhealthy alcohol use. | Unhealthy alcohol use a method to avoid HIV status disclosure |
| Lack of family and social support | Not measured | People with unhealthy alcohol use perceived that their families shunned them and they did not offer moral support as needed |
| Those smoking cigarettes were also addicted to unhealthy alcohol use | Men (AOR, 0.41; 95% CI, 0.23–0.72) and baseline smokers (AOR, 0.49; 95% CI, 0.25–0.96) also had lower odds of reducing their alcohol use after ART commencement. There was also increased odds of alcohol reduction at Clinic B (AOR, 2.91; 95% CI, 1.695.04) compared to Clinic A. | Smokers more likely to participate in unhealthy alcohol use |
| **Healthcare Environment (HIV health system, clinic, and provider factors)** | | |
| Lack of alcohol reduction support at HIV clinics | Reduced unhealthy alcohol use was nearly 3 times more common at Clinic B compared to Clinic A (AOR = 2.91, CI 1.69–5.04), suggesting service variability and/or impact. | Insufficient and inconsistent alcohol reduction support at clinics. Support services vary among clinics. |
| Insufficient training of HCW in alcohol consumption assessment and treatment in PLWHA | During the first year on ART the overall prevalence of unhealthy alcohol use in the analysis group reduced significantly from 40.4% to 29.6% (P<0.01). Of the 280 with unhealthy use at baseline, 122 (43.6%) reported a lower degree of alcohol consumption at 1 year. | Little support is available at the community level. Lack of HCW training in alcohol reduction. |
| Inconsistent counseling/ messaging at HIV clinics | Increased odds of alcohol reduction at Clinic B (AOR, 2.91; 95% CI, 1.695.04) compared to Clinic A. | HCW provision of mixed messages to patients (reduction to safe levels vs. abstinence). |

Many HCWs reported insufficient training in assessment and counselling of alcohol consumption in their patients on ART that resulted in failure of standard of care counseling to reduce alcohol intake among PLWHA.

## Discussion

At two public sector clinics in urban Zambia, we documented a high prevalence of unhealthy alcohol use among adult men and women receiving HIV care [9]. Moderate reduction in unhealthy alcohol use occurred post-ART initiation; however, nearly one-quarter of the cohort had persistent unhealthy alcohol use. Our data also revealed notable heterogeneity in alcohol reduction at the patient and facility levels. To better understand quantitative data on patterns of alcohol use in PLWHA, we conducted an explanatory qualitative sub study. This revealed that persistent unhealthy alcohol use may be due to unaddressed motivators and mechanisms, as well as low capacity of HCWs to respond to the problem of unhealthy alcohol use [1–5]. Combined analysis of quantitative and qualitative findings showed that addressing unhealthy alcohol use among PLWHA will require capacity building among HIV clinic staff and consideration of patient sociodemographic and structural factors as well stigma [5, 7].

In our quantitative data, alcohol use declined after ART start, which may be due to clinical care received. Although we did not measure the type and duration of counseling received by each participant, our finding that nearly 40% of people with unhealthy alcohol use had moderate or no use at 1 year should be seen as evidence that current approaches work for some patients. Decline in alcohol use after ART is supported by other research [19] but one report from Uganda, which used biomarkers to supplement self-reports found increasing alcohol use post-ART start [20]. Currently, only brief, and unstructured interventions are widely available at HIV clinics in Zambia [1]. Alcohol brief interventions are recommended as they are easy to provide; however, several rigorous trials in PLWH who reported unhealthy alcohol use failed to show a benefit of BIs compared to standard care [21, 22].

Our quantitative data revealed that women, people who did not smoke, and those attending Clinic B were more likely to reduce their alcohol intake compared. Nearly 1 in 4 women reported unhealthy alcohol use at ART start, which is higher than many other studies in the region. In qualitative data, it was reported that alcohol use by women has become increasingly normative in the past few decades, especially in younger women. However, compared to men, women were perceived to have more of a restraint on their alcohol indulging habits due to their child-rearing responsibilities. This could explain their propensity to reduce their intake after ART start. Sex differences in alcohol use patterns are supported by an analysis of PLWHA on 3 continents, women were less likely to drink than men in Uganda but similar in Russia and the United States. Smoking is a common co-occurring substance used with alcohol [23]. Our finding that people who smoked had lower odds of reducing their alcohol intake could be because they had higher baseline alcohol use, which could not be discerned with the AUDIT-C. Finally, we were surprised to see clinic level difference in alcohol reduction since the two clinics involved in the study had similar characteristics. That patients with baseline unhealthy alcohol use at Clinic B had nearly 3 times the odds of alcohol reduction compared to their counterparts at Clinic A will require additional analysis. This variation could be due to 'software' factors, such as the emphasis placed on alcohol use diagnosis and treatment by clinic leadership or staff capacity. Such variation in health outcomes was also noted in Zambian HIV clinics in a study of HIV-related mortality.

While not measured quantitatively, other patient-level factors associated with failure to reduced one's alcohol use reported in qualitative data included untreated mental health symptoms, unemployment, and the simple pleasure of drinking [14, 15]. Such factors also emerged as important in a study in Uganda [24]. Mental health comorbidity is common among

PLWHA who report unhealthy alcohol use and these comorbidities also undermine HIV care [1, 24]. At present, most alcohol-focused treatments do not address these comorbidities within the same protocol [1].

Another major finding from this study was the influence of the healthcare environment on alcohol use outcomes. HCW reported low perceived competency to manage PLWHA who drank heavily. HCW reported not knowing the exact definition of unhealthy alcohol use in the Zambian context, especially as beer was seen as a traditional beverage, and lacking the appropriate guidance to determine if PLWHA have unhealthy alcohol use. Participants reported mixed experiences discussing alcohol use with HCWs, which likely contributed to whether they reduced their alcohol use. This builds on other data [24]. Major gaps in training of HIV service providers in substance use and mental health interventions exist [25]. Difference in alcohol reduction by site was likely due to interactions between the hardware (staff levels, space) and software (competency, commitment to issue, etc.) of the health system at different sites, something that has been described for HIV/TB care previously in Zambia [26].

We also documented intersectional stigma, the convergence of multiple stigmatized identities, between living with HIV and having unhealthy alcohol use. Both are stigmatized in Zambian society and people with HIV who are also labeled as unhealthy alcohol users may experience enhanced levels of stigma from health workers and other community members. It has been documented that people with HIV on ART are unlikely to openly report low adherence for fear of 'being punished' through referral to enhanced adherence counseling, where patients are given only small dispensations of ART at a time until they demonstrate good behavior. Less is known how alcohol-related stigma may impact screening and treatment of unhealthy alcohol use in this context.

## Strengths & limitations

There were several major strengths to this study. The strengths of this study include the real world and prospective longitudinal nature of the HIV cohort; most previously reported HIV-alcohol studies in SSA were cross-sectional. Another strength was our integration of qualitative and quantitative data to understand longitudinal alcohol use patterns. While rural settings and other countries may have a different degree of unhealthy alcohol use, it is likely that much of the data generated in this study is relevant in other settings in SSA where alcohol poses a barrier to addressing the HIV epidemic.

While our analysis had strengths, there were also limitations. First, we used a non-randomized before-after design; hence, patterns of alcohol use might not be caused by receipt of HIV care, which included a degree of counseling, but rather by other factors. Secondly, we did not characterize (number of sessions, duration, content) the standard care alcohol counseling that was provided, and it is probable that counseling was inconsistent. Therefore, we are unable to comment on dose response or whether some clients ever were counseled based on their AUDIT-C score. Self-reports by clients may be biased by their inability to recall their alcohol intake, and their possible fear of judgement by the HCW increasing the difficulty to interpret the differences in outcomes between the two clinics. Thus, reported alcohol reduction may have been inaccurate. Further, we recruited consecutive participants meeting cohort criteria; however, it is possible that those with unhealthy alcohol use or more severe illness were less likely to have been enrolled, which could reduce the generalizability of our findings.

## Conclusion

Unhealthy alcohol use was widespread among PLWHA who initiated ART in urban Zambia. After ART start there was an overall moderate reduction of alcohol use; however, nearly 1 in 4

had persistent unhealthy alcohol use possibly due to unaddressed mechanisms and motivators to drink and low clinic staff capacity to recognize and address alcohol use in patients. Integrated, scalable, and evidence-based approaches to screening for and managing unhealthy alcohol use are needed in SSA to help address alcohol's negative impact on the HIV epidemic.

## Supporting information

**S1 File. Focus group guide for community members (Bemba).**
(PDF)

**S2 File. In-depth interview guide for community members (Bemba).**
(PDF)

**S3 File. Focus group guide for community members (English).**
(PDF)

**S4 File. In-depth interview guide for community members (English).**
(PDF)

**S5 File. Focus group guide for community members (Nyanja).**
(PDF)

**S6 File. In-depth interview guide for community members (Nyanja).**
(PDF)

**S7 File. In-depth interview guide for health care workers.**
(PDF)

## Author Contributions

**Conceptualization:** Helen Smith, Michael J. Vinikoor.

**Data curation:** Anna Helova, Helen Smith, Michael J. Vinikoor.

**Formal analysis:** Anna Helova, Helen Smith.

**Methodology:** Anna Helova, Michael J. Vinikoor.

**Project administration:** Helen Smith.

**Supervision:** Michael J. Vinikoor.

**Writing – original draft:** Mah Asombang, Anna Helova, Jenala Chipungu, Anjali Sharma, Gilles Wandeler, Jeremy C. Kane, Janet M. Turan, Helen Smith, Michael J. Vinikoor.

**Writing – review & editing:** Mah Asombang, Anna Helova, Jenala Chipungu, Anjali Sharma, Gilles Wandeler, Jeremy C. Kane, Janet M. Turan, Helen Smith, Michael J. Vinikoor.

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
