## [Decision Letter · Decision Letter 0]

4 Oct 2021

PGPH-D-21-00538

Alcohol reduction outcomes following brief counseling among adults with HIV in Zambia: A sequential mixed methods study

Dear Dr. Asombang, 

Thank you for submitting your manuscript to PLOS Global Public Health. After careful consideration, we feel that it has merit but does not fully meet PLOS Global Public Health’s publication criteria as it currently stands. Therefore, we invite you to submit a revised version of the manuscript that addresses the points raised during the review process. In the revision, make sure to address the reviewers' concerns about the methodology, interpretation of the data and presentation of the paper.  

We look forward to receiving your revised manuscript.

Kind regards,

Roopa Shivashankar, MD, MSc

Academic Editor

Journal Requirements:

1. Please include a copy of the interview guide used in the study, in both the original language and English, as Supporting Information, or include a citation if it has been published previously.

2. Please update the completed 'Competing Interests' statement, including any COIs declared by your co-authors. If you have no competing interests to declare, please state "The authors have declared that no competing interests exist". Otherwise please declare all competing interests beginning with the statement "I have read the journal's policy and the authors of this manuscript have the following competing interests:"

3. Please remove the figures embedded in your manuscript file leaving only the separate figures files.

4. Since your data is not available for proprietary reasons, please explain via email why the data is not available. Please also include the contact information for the third party organization that should be contacted should other researchers want to request access to this data and please include the full citation of where the data can be found. We also request that you verify with us via email that any researcher will be able to obtain the data set in the same manner that the you have obtained it. If you feel you are unwilling or unable to adhere to this policy, please explain your reasons by return email and your exemption request will be escalated to the editor for approval. Your exemption request will be handled independently and will not hold up the peer review process, but will need to be resolved should your manuscript be accepted for publication. One of the Editorial team will be in touch if they require more information.

5. Please amend your detailed Financial Disclosure statement. This is published with the article, therefore should be completed in full sentences and contain the exact wording you wish to be 

i) State the initials, alongside each funding source, of each author to receive each grant.

iii). State what role the funders took in the study. If the funders had no role in your study, please state: “The funders had no role in study design, data collection and analysis, decision to publish, or preparation of the manuscript.”

Reviewers' comments:

Reviewer's Responses to Questions

**Comments to the Author**

1. Does this manuscript meet PLOS Global Public Health’s publication criteria? Is the manuscript technically sound, and do the data support the conclusions? The manuscript must describe methodologically and ethically rigorous research with conclusions that are appropriately drawn based on the data presented.

Reviewer #1: Yes

Reviewer #2: Yes

2. Has the statistical analysis been performed appropriately and rigorously?

Reviewer #1: Yes

Reviewer #2: Yes

3. Have the authors made all data underlying the findings in their manuscript fully available (please refer to the Data Availability Statement at the start of the manuscript PDF file)?

Reviewer #1: No

Reviewer #2: No

4. Is the manuscript presented in an intelligible fashion and written in standard English?

Reviewer #1: Yes

Reviewer #2: No

5. Review Comments to the Author

Reviewer #1: This is an informative and relevant study and the findings can potentially have important uses. Please see my comments below.

• Was the 3-question tool used or the 10-question Audit C tool used?

• Page 7, quantitative methods: What is the variable ‘clinic’ used in regression?

• Page 8, qualitative methods: The IDIs were conducted with individuals who self-reported reduced use and persistence use: self-reports are often not very reliable, especially in alcohol users; why did the investigators not objectively assess change in alcohol use over one-year period, since these individuals were being followed-up as part of the study?

• Page 8, qualitative methods: Anderson`s Behaviour Model (ABM) is supposed to be used as a framework to design the methods as well as interpret the results. While the authors have interpreted the results using ABM, please describe how the model was used as a framework in designing the methods.

• Page 8, qualitative methods: Please give more details about the thematic analysis approach.

• Page 8, quantitative results: Was the sample size statistically calculated? If yes, please give details. If no, please justify the sample size.

• Page 8, quantitative results: Consecutive enrolment was used as reported by the authors; this raises the question of how representative the study population was of the general population. In such a method, it is likely that heavy drinkers or PLWHs with severe illness may not participate in the study and thus an important stratum of alcoholic PLWHs may have been left out of the study. This should be mentioned as a study limitation.

• Page 8, quantitative results: The Audit C tool scores are interpreted as low, increasing and high risk and dependence. There are not reported in the manuscript. Further, what is the definition of unhealthy alcohol use and moderate alcohol use and how have the authors calculated these? The authors should maintain consistency between methods and results.

• Page 9, quantitative results: Regression analysis should adjust for some measure of limited alcohol-reduction counselling, i.e., whether given or not, how many times, etc., since this variability in this intervention can potentially alter alcohol consumption practices on its own. Also, the authors state that one of the conclusions from this study is that current counselling practices are inadequate. However, without knowing about the details of this counselling and accounting for this counselling in analysis, such a conclusion cannot be made. While the authors have mentioned this as a limitation, is it possible to go back, search for data regarding this variable and redo the regression analysis? If this is not possible, the authors should remove this sentence from conclusion, discussion and abstract sections.

• Page 16, Discussion: Results should always be interpreted and discussed with caution in mixed-methods study. Because qualitative results can usually not be generalised to the larger population, combining these with quantitative results, which can be generalised, gives an impression that the qualitative results are universally applicable. To avoid this, in the discussion section, qualitative results should be specifically marked as qualitative. For e.g. the authors state that standard of care counseling appeared to fail to reduce alcohol intake, which looks like an universal statement. However, this variable was not analysed in quantitative results and this statement is a result of qualitative data analysis and the same should be stated.

Reviewer #2: The findings from the manuscript are useful. However, major revision is needed. please refer to the points below:

1. Usually PLWHA is used for “people living with HIV/AIDS”

2. All the references need to be rewritten and rechecked as there is no uniformity. Moreover, certain references, e.g., 15 are incomplete.

3. References need to be re-checked. In the “Introduction”, after 5, immediately 11th reference is coming… the in-between numbers are scattered in the methodology.

4. Introduction 2nd paragraph – “Despite its potentially negative impact on the HIV care continuum, unhealthy alcohol use remains largely unaddressed, as evidence-based interventions to reduce alcohol use are rarely implemented in Zambian hospital and clinical facilities (11,12,14) – the studies mentioned are talking about AUD and PLWH; not specifically about Zambia. Thus, the sentence or reference need to be modified accordingly.

5. Quantitative section – what was the need for “second enrolment”? Did the initial ethical approval include drawing of blood samples?

6. Focus Group Discussions is abbreviated as “FGD’s”. Ethical permission obtained for qualitative analysis?

7. Methodology section needs to be made more crisp, exclude unnecessary details on how people are counselled at the center etc.

8. The statement before table 1 – “Unhealthy alcohol use was reported at baseline by 280 (40.4%) patients including 26.7% of women and 57.0% of men (P<0.01) and the figures given in table 1 do not match.

9. In qualitative results, what is WLWH/MLWH/ARVs?

10. What is the difference between clinic A and B – in terms of staff strength, characteristics, treatment or counselling available etc.?

11. “UNAIDS 95-95-95 “ - reference

12. Discussion is haphazard and needs to be re-written.

6. PLOS authors have the option to publish the peer review history of their article (what does this mean?). If published, this will include your full peer review and any attached files.

**Do you want your identity to be public for this peer review?** For information about this choice, including consent withdrawal, please see our Privacy Policy.

Reviewer #1: No

Reviewer #2: **Yes: **Swati Kedia gupta

---

## [Decision Letter · Decision Letter 1]

5 Jan 2022

PGPH-D-21-00538R1

Alcohol reduction outcomes following brief counseling among adults with HIV in Zambia: A sequential mixed methods study

Dear Dr. Asombang,

Thank you for submitting your manuscript to PLOS Global Public Health. After careful consideration, we feel that it has merit but does not fully meet PLOS Global Public Health’s publication criteria as it currently stands. Therefore, we invite you to submit a revised version of the manuscript that addresses the points raised during the review process.

The reviewers comments have been adequately addressed. There are still a couple of minor suggestions at the end of this mail.  The manuscript still lacks the clarity of scientific writing in many places. The academic writing requires clarity and therefore avoid using words that are commonly spoken but grammatically incorrect.  Some examples are below but it is not an exhaustive list. Please check the complete manuscript for language clarity and grammer errors. 

**Abstract **

 Data from sub-Saharan Africa are limited on the impact of alcohol on the HIV epidemic, particularly related to longitudinal responses of people living with HIV (PLWHA) to clinical care that includes counseling on alcohol reduction.

The sentence is unclear. Use shorter sentences.

Among 693 participants analyzed.

Replace with "Among the data of 693 participants". We do not analyse people but the data. 

**Methods**

treating unhealthy alcohol integrated with HIV services.

Replace with "treating unhealthy alcohol use integrated with HIV service"

When alcohol was reported, this counseling included brief and unstructured information on  alcohol reduction 

Replace with "When alcohol use was reported, this counseling included brief and unstructured information on  reducing alcohol consumption".  Alcohol reduction may mean alcohol content in a drink or body is reduced. 

We defined unhealthy alcohol use at AUDIT-C score of 3-12 for women and 4-12 for men (15), based on World Health Organization (WHO) guidance.

Replace with " We defined unhealthy alcohol use if a participant had an AUDIT-C score of 3-12 for women and 4-12 for men (15), based on World Health Organization (WHO) recommendation"

Moderate use was 1-2 points for women and 1-3 for men and abstinence was 0 points

Was this was based on WHO recommendation?

Ensure to spell out all numbers <10. 

We look forward to receiving your revised manuscript.

Kind regards,

Roopa Shivashankar, MD, MSc

Academic Editor

Journal Requirements:

1. We have noticed that you have uploaded supporting information but you have not included a list of legends.  Please add a full list of legends for all supporting information files (including figures, table and data files) after the references list.

Additional Editor Comments (if provided):

Reviewers' comments:

Reviewer's Responses to Questions

**Comments to the Author**

1. If the authors have adequately addressed your comments raised in a previous round of review and you feel that this manuscript is now acceptable for publication, you may indicate that here to bypass the “Comments to the Author” section, enter your conflict of interest statement in the “Confidential to Editor” section, and submit your "Accept" recommendation.

Reviewer #1: All comments have been addressed

Reviewer #2: All comments have been addressed

2. Does this manuscript meet PLOS Global Public Health’s publication criteria? Is the manuscript technically sound, and do the data support the conclusions? The manuscript must describe methodologically and ethically rigorous research with conclusions that are appropriately drawn based on the data presented.

Reviewer #1: Yes

Reviewer #2: Yes

3. Has the statistical analysis been performed appropriately and rigorously?

Reviewer #1: Yes

Reviewer #2: Yes

4. Have the authors made all data underlying the findings in their manuscript fully available (please refer to the Data Availability Statement at the start of the manuscript PDF file)?

Reviewer #1: Yes

Reviewer #2: No

5. Is the manuscript presented in an intelligible fashion and written in standard English?

Reviewer #1: Yes

Reviewer #2: Yes

6. Review Comments to the Author

Reviewer #1: None

Reviewer #2: Minor corrections

1. In qualitative section. Utujilijili = tujilijili

2. References are not yet in one standard format. It would be good if a referencing system, eg., zotero is used

3. The table for integrating qualitative and quantitative results needs formatting as the results are getting spread across rows.

7. PLOS authors have the option to publish the peer review history of their article (what does this mean?). If published, this will include your full peer review and any attached files.

**Do you want your identity to be public for this peer review?** For information about this choice, including consent withdrawal, please see our Privacy Policy.

Reviewer #1: No

Reviewer #2: **Yes: **Dr Swati Kedia gupta

---

## [Editor Report · Decision Letter 2]

2 Feb 2022

Alcohol reduction outcomes following brief counseling among adults with HIV in Zambia: A sequential mixed methods study

PGPH-D-21-00538R2

Dear Dr Asombang,

We are pleased to inform you that your manuscript 'Alcohol reduction outcomes following brief counseling among adults with HIV in Zambia: A sequential mixed methods study' has been provisionally accepted for publication in PLOS Global Public Health.

Best regards,

Roopa Shivashankar, MD, MSc

Academic Editor